# Degradation Analyses of Systemic Large-Panel Buildings Using Comparative Testing during Demolition

**DOI:** 10.3390/ma15113770

**Published:** 2022-05-25

**Authors:** Maciej Wardach, Janusz R. Krentowski, Piotr Knyziak

**Affiliations:** 1Faculty of Civil Engineering and Environmental Sciences, Bialystok University of Technology, Wiejska 45E, 15-351 Bialystok, Poland; janusz@delta-av.com.pl; 2Faculty of Civil Engineering, Warsaw University of Technology, Politechniki sqr. 1, 00-661 Warsaw, Poland; pk@il.pw.edu.pl

**Keywords:** large-panel, degradation, NDT, destructive testing, demolition

## Abstract

Assessment of the technical condition of large-panel buildings, due to their on-going use and covering resulting from thermomodernization works, is problematic. Results from non-destructive tests (NDT) are subjected to high uncertainty. Destructive tests, which give results with the highest level of confidence, are practically not used. Local sampling for testing gives only a partial image of the condition of a prefabricated building. In this type of construction, joints connecting the precast elements are the most vulnerable to degradation. Access to them is technically difficult. Demolition of this type of building is extremely rare. However, it is a unique opportunity to perform a full spectrum of both NDT and destructive testing. This gives an opportunity for large-scale demolition sampling to identify hidden defects and compare the results obtained by different methods. The comparison of results allows for the scaling of NDT methods and reveals the presence of typical relationships. The paper presents visual, non-destructive, and destructive tests’ results of an over 40-year-old large-panel building scheduled for demolition. The design of this building is repetitive and similar to solutions found in thousands of other buildings. The usefulness of particular research methods for evaluating the technical condition of prefabricated buildings has been determined.

## 1. Introduction

Reinforced concrete structures are influenced by various factors. Their destructive impact is important for a building’s durability [1,2,3].

Prefabricated buildings were commonly constructed in many European countries be-tween 1970 and 1990 [4,5,6,7,8]. Prefabricated construction was characterized by using typical wall and floor elements and repetitive system solutions. Thousands of buildings were constructed using similar or even identical solutions of prefabricates and their connections in the structure. This type of construction allows the whole group of system buildings to be evaluated through numerically limited analyses. For destructive testing, even individual cases provide valuable research material. A wide range of tests performed during the demolition of the building allows to obtain comprehensive results.

The technical condition of structural elements made of reinforced concrete prefabricates depends on many factors. The most important ones are the quality control of prefabricates in the production plants, the way of transport and storage on the construction site, the accuracy of their assembly, the quality of filling the joints with concrete and, in the next stage, the proper maintenance of the building [9]. Natural processes of ageing of materials, especially those exposed to the influence of an aggressive environment, are nevertheless important. In terms of durability of the structure of large-panel buildings, the potentially weak points are the vertical and horizontal joints connecting the individual prefabricated elements. Moisture penetrating through the exterior wall elements propagates through the joints to the interior wall sections. Water from rooms that generate moisture (kitchens, and bathrooms) penetrates the joints located directly adjacent to them. External elements made of prefabricated elements, i.e., walls, flat roofs, balconies, and loggias, are directly exposed to the influence of moisture, temperature and pollutants present in the environment.

In the studied type of buildings, a typical prefabricated system was used for external curtain walls. It consisted of a concrete structural layer, thermal insulation from expanded polystyrene or mineral wool, and an external layer of textured concrete [8,10]. The durability of the external partition formed in this way is determined by the effectiveness of the implementation of the connection of the concrete layers. Examples of degradation of these types of elements are widely discussed in the literature [10,11,12,13,14].

The quality of the workmanship in these types of buildings during construction and after commissioning raised doubts regarding their durability, mainly in terms of corrosion protection. During several decades of use, the condition of both prefabricated elements and connections between elements was gradually degrading. Structural elements, as well as whole buildings, required regular repairs or modernization [15,16,17,18,19,20]. Secondary elements, such as flashings or balcony railings, were also subject to gradual degradation [11,21]. System changes in Central Europe negatively influenced the process of utilization. Many repairs were delayed for too long for financial reasons.

Changes in energy costs and regulations concerning the limit values for heat transmission coefficients and thermal insulation of buildings have occurred in various countries. This resulted in the need to insulate external walls [22,23,24,25]. However, the addition of new continuous surface layers makes it impossible to monitor the surface condition of structural elements and their connections from the external side. On the inside, ongoing maintenance of dwellings, repairs, and surface layers of paint added by owners also make it difficult to detect anomalies and correctly assess the extent of the processes taking place.

Almost all large-panel buildings that have been constructed over the past few decades are still in use. Most of them are large multi-staircase and multi-storey residential buildings, but also office buildings, hotels, and public buildings. Buildings require condition assessment and ongoing monitoring for defects. Proper assessment also requires inspections inside dwellings. Such surveys are troublesome for residents, so they are rarely performed. The results of non-targeted scientific research on a group of buildings—using destructive [26] and only visual [16] tests—can be used as comparative material. The inspection is limited to parts of public spaces, staircases, basements, elevations, and roofs. In the literature, the methods of diagnosis and modernization of large-panel buildings have been widely described. In the past, due to the specificity of local and national systems, they were mostly European publications [7,27,28]. In recent years, there are more and more publications showing a wide range of problems and solutions applicable to most of the large-panel systems in different countries [29,30].

Despite thousands of prefabricated buildings, their demolition is a rarity and an extremely valuable research field. During the demolition of many unnecessary buildings in the former German Democratic Republic, most probably no detailed studies were performed. It was either a purely commercial activity or the results were published in the literature with limited scope. Mainly aesthetic and heat saving aspects were previously analysed [31]. On the structural side, there is one available publication on the case of demolition of a large-panel building that was damaged due to underground mining [32]. Buildings remaining in an unfinished state have also been a testing ground, with a smaller scope of research [33,34]. An interesting study was conducted on buildings after the 2019 earthquake in Albania [35].

The present development of testing methods and equipment makes it possible to carry out more and more non-destructive tests. Sclerometric [36,37,38], ultrasound [39,40,41,42,43], ferromagnetic [44,45], and thermovision [46] tests, which are widely described in the literature, are most commonly used for testing reinforced concrete structures. An interesting comparison of testing methods in the aspect of concrete strength assessment has been published in [47].

Technological innovations are also revolutionizing laboratory research, such as Digital Image Correlation (DIC), which allows for non-contact determination of selected concrete fracture mechanics parameters [48,49,50]. Cracks are one of the most common signs of concrete degradation that cause concern to users. For the investigation and analysis of plastic shrinkage cracks, microCT scanning, which has been widely described in [51], is used. In the literature, an interesting application of intelligent algorithms enabling autonomous crack detection using a digital image processing system has been presented [52]. Cracks, apart from affecting the visual aspects, may influence the load-bearing capacity of the elements. Crack growth is particularly undesirable in elements exposed to aggressive environmental influences. The penetration of chlorides into the concrete can significantly accelerate the degradation process. The diffusion properties of cracked concrete as a function of crack width have been widely reported in the literature [53], and the chloride diffusion coefficient for degraded concrete has also been investigated [54]. In terms of numerical analyses, a three-phase 3D computational model was developed to simulate chloride diffusion in concrete [55]. The progress of technology and numerous scientific studies allow for a better understanding of the fracture mechanics of concrete, which makes it possible to predict the course and depth of cracks.

Results obtained by non-destructive methods, due to economic and technical aspects, are rarely confirmed by a large number of destructive tests carried out on existing degraded objects.

Demolished buildings are not only valuable research objects but also a source of materials that can be recycled. The construction of large-panel buildings was based on the use of large quantities of concrete. This provides an opportunity to recover aggregate, which can be successfully used to produce new structural elements. This is an increasingly common process in construction practice. Recycled aggregates are currently the subject of extensive scientific research. Results of research on combining recycled aggregates with geopolymer concrete are particularly interesting [56]. Intriguing studies on the reinforcement of recycled aggregates with pozzolanic slurries have been described in the literature [57]. Research has been conducted into innovative applications for these aggregates, which include high-temperature resistant hollow blocks, as widely described in [58]. Demolition of a previously used building can also be a source of rubber waste. The results of analyses related to the use of this waste for concrete production were presented in [59]. Nowadays, the use of eco-friendly solutions throughout the construction process is highly desirable. In addition to materials from demolition, waste from the natural environment can also be used in the construction industry. An example of this would be the seashells that have been tested for use in the production of cementitious materials, as described in [60,61]. Reducing the amount of solid waste as well as the consumption of primary raw materials is an increasingly common theme in research papers. This contributes to the spread of eco-friendly solutions in the construction industry.

The authors of the present study participated in the demolition of a 12-storey building (Figure 1), where they performed a number of visual, destructive, and non-destructive tests. The obtained results and formulated conclusions may contribute to the improvement of large-panel buildings technical condition assessment. Particularly valuable is the comparison of the results of visual and non-destructive tests with the results of destructive tests, which allows calibration of assessments.

This paper aims to determine the suitability of particular testing methods for assessing the degradation state of structural elements of prefabricated objects. The places where destructive testing is necessary were located. Hidden defects, impossible to detect with currently used testing apparatus, were identified.

## 2. Investigations

The demolished two-segment building with 11 overground storeys and one underground storey was constructed in 1978 using OWT-67 technology (a prefabricated large-panel building, one of the main types of large-panel building systems in Poland) and served as an office building. The system was characterised by the size of the largest module being 5.4 × 4.8 m, with a storey height of 2.7 m [8]. The 0.14 m thick floor slabs were supported by three walls, also 0.14 m thick, and an external beam-wall. The walls were made of slabs of the height of a storey and the length of a room. The façade of the described office building consisted of plates attached to the gable walls and beams. There were also two one-storey parts adjacent to the building of mixed, reinforced concrete and steel construction, which constituted the main entrance to the entire complex. The building, which was decommissioned in 2011, was demolished due to architectural and economic aspects rather than damage and structural defects.

### 2.1. Demolition Technology

Due to the building’s location in the city centre and its proximity to the adjacent building, it was decided to demolish in a way that minimized the inconvenience to neighbouring buildings and their occupants. The use of explosives is characterized by considerable dust, vibration effects on the surroundings, enormous noise, as well as specific requirements regarding the size of the construction site and type of structure. Ultimately, mechanical demolition using demolition excavators (Figure 2b,c) and light equipment in the form of drill hammers and steel circular saws operated by skilled manual workers (Figure 2a) was decided upon. Working in this manner allowed the authors to monitor the ongoing progress of the work and enabled the collection of samples for laboratory analysis and assessment of structural degradation from each stage of demolition.

### 2.2. Non-Destructive Tests

Prior to demolition works, visual assessment of the technical condition of the building was carried out. The assessment was conducted in the context of safety of people performing the survey and dismantling works, but also in the context of visible damage and signs of the building’s destruction. The survey was performed in accordance with the typical methodology of periodic inspections of buildings. At this stage of the study, there were no significant visible signs of degradation threatening the structure, which could raise suspicion of safety hazard during demolition. Also, the visible signs were not indicative of significant threats to the durability of the structure if it continued to be in use.

In the horizontal joints of the floor slabs, cracks were visible. They were the evidence of vertical displacement of the edges of the adjacent slabs. In the upper parts of vertical joints of prefabricated walls, it was found that the fillings made during the building assembly were made without adequate precision. The concrete was very porous with an uneven surface and was not filling the entire wall joint. In the beam-walls, corroded rebar was inventoried where the window and door frames were supported (ref. Figure 7a). The corrosion could have been caused by a careless demolition of the window sills. During removal of window joinery, the workers damaged concrete layer and exposed reinforcing steel to environmental impacts. Elsewhere, no discoloration or cracks along the bar mesh, which could indicate an intensive corrosion process of the reinforcement bars, were found.

For non-destructive testing, the authors used specialized testing equipment in the form of ultrasound (Figure 3a,b) and ferromagnetic (Figure 3c) methods.

Non-destructive testing methods for the diagnosis of concrete structures have been widely described in the literature [62,63,64]. Their limitations and possibilities of supplementing the results by combining individual methods have also been formulated [65]. Taking into account the wide availability of methods’ descriptions in the literature, they have been omitted in this case, and only the main parameters of the equipment used in the conducted tests are presented.

In the first stage, linear scans of longitudinal and transverse walls were made using ultrasound equipment with 54 kHz heads. Two-way access to the elements was used, thanks to doorways and openings created after excavations and core drilling. Five walls on each of floors −1, 3, 5, and 10 were examined. The diameter of the longitudinal wave transducer was 5 cm. The measurement points were marked with wax chalk at 10–15 cm intervals. Considering the dimensions of the transducers, in reality, the spacing between the edges of the heads was between 5 to 10 cm. A measurement grid with this spacing allows the surface under examination to be scanned accurately. Defects that could hide between the measuring points are negligibly small and would not affect the load-bearing capacity of the component. A total of 5 to 10 readings were taken on each wall. The number of readings was due to difficult access and the limited length of the cables feeding the transducers (testing with two-way access was only possible in walls with openings). Wave propagation velocities for all readings taken ranged from 3500 to 4000 m/s. According to the classification presented in [66], this indicates good concrete quality. The results obtained during the testing of the 5-storey transverse wall, along with two graphs of wave propagation in concrete are presented in Figure 4.

The next stage was surface scans of walls and floors using a Pulse-Echo head. The thickness of the elements and the location of any voids and material discontinuities, as well as the propagation velocities of the shear waves, were analysed. The measurement grid ranged from 50 × 50 to 50 × 100 cm, with intervals of 10 cm. The thicknesses of the elements were within the execution deviations, i.e., ±0.5 cm. No areas indicating the presence of material discontinuities were located on the obtained images. The transverse wave velocity was within the range of 1800–2000 m/s, while the average wave velocity for concrete is usually 2000 m/s [67]. Images of post-surface scans for one of the 3rd floor walls are presented in Figure 5.

Due to one-sided access to the vertical joints, the concrete quality was checked using indirect measurement mode, i.e., setting the heads on one surface. The tests were performed using 54 kHz transducers. In order to eliminate the uncertainty related to the length of the measurement path, one of the device functions was used. It allows to take readings by moving one of the heads while keeping the other one stationary. After taking 4 readings, with known path length, the device is able to draw a curve and estimate the longitudinal wave velocity. The obtained velocities ranged from 1700 to 2100 m/s. This may be an indication of poor concrete quality, but it should be noted that in surface measurement mode, the quality of the top layer of the element has a significant influence on the results. Micro-cracks and defects lead to under-estimation of wave velocity.

The next stage was to perform B-scans of vertical joints and walls using the Pulse-Echo head. B-scan generates an image of the cross-section of the tested element, perpendicular to the scanning surface. The defects and discontinuities in the presented results are colour-coded from pink to purple. Heterogeneities in the concrete affect the propagation of the ultrasound pulse. They cause scattering of the signal coming from the Pulse-Echo head. The device measures the transit time of the wave and its amplitude. The colour in the image becomes darker as the amplitude of the wave increases. The local maximum of the amplitude results from the reflection of the wave at the boundary between the concrete and the air filling the structural defect. In the case of the darkest colour (purple), the wave amplitude was the highest. This indicates that the wave encountered significant material heterogeneity in the form of voids or delaminations. B-scans performed revealed the presence of voids and delaminations in the fillings (Figure 6a), which were caused by incorrect placement and compaction of the concrete mix during construction of the elements. In the walls, no irregularities indicating poor concrete structure were detected (Figure 6b).

In order to identify the quality of the reinforcement work, line scans (Figure 7c) and area scans were performed using ferromagnetic testing equipment. Line scanning allows to check the quality of the reinforcement work along one line. This measurement mode can be used for scanning beam elements or columns, while area scans should be used for surface elements such as walls and slabs. Measurement is performed on the basis of a defined measurement grid. Area scanning makes it possible to quickly determine the distribution of the reinforcement and the thickness of the coverings over large areas, where the distribution of rebars can be highly variable. Excavations were made at the non-destructive test locations to compare the results.

The results of the scans indicated insufficient cover thickness locally. The reinforcement distribution was consistent with the available original documentation.

Using the sclerometric method, the concrete class of the precast elements was estimated based on the reflection number. It was determined that it is equivalent to the current class C16/20 according to EC2 [68].

Locally, scratches were observed in the walls above the door openings. The cracks were measured with a Brinell magnifying glass and ranged from 0.3 to 1.1 mm. An ultrasound method was used to identify the depth of the cracks. The longitudinal wave transducers were placed on both sides of the crack. First, each transducer was placed at a distance of *a* = 100 mm from the crack, after which the ultrasound wave transit time was measured. The transducers were then moved apart to a distance of 2*a* = 200 mm and a second reading was taken. Based on the difference in transit times obtained from the two readings, the crack penetration depth was estimated. Debris was removed from the crack using an industrial vacuum cleaner. Cleaning is very important in penetration measurements using the ultrasound method, because wave propagation through the contamination can significantly under-estimate the readings relative to the actual condition. The depth of penetration was determined to be between 20 and 65 mm. An example of crack depth for a longitudinal wall and a transverse wall is shown in Figure 8.

### 2.3. Destructive Tests

The accuracy of the non-destructive testing results was continuously verified by performing a large number of wall (Figure 7b), beam-wall and floor (Figure 9b) excavations. The building was scheduled for demolition, so the only limitation of the testing was to maintain structural integrity. The rebar exposed in the excavations showed no signs of corrosion. The concrete in the precast elements was of good quality and had no high porosity. Attention was paid to the large dimensions of the aggregate used to make the elements, exceeding as much as 40 mm in places. The thickness of the floor screeds ranged from 35 to 50 mm. Numerous explorations have not revealed excessive thickness values of surface layers added during renovations, which could lead to excessive slab deflection and stippling of the partitions [69].

In order to perform laboratory tests, core drillings were made in different parts of the structure (Figure 9a). The locations and diameters of the drill holes were selected in accordance with EN 13,791 [70]. In the beam-wall elements, the thickness of individual layers was measured in situ. The thickness of the insulation was found to be 5 cm, while the texture layer was characterized by different thickness values, ranging from 3.5 to 7.5 cm, which is inconsistent with the design specification. The differences were not visible from the outside because of the facade made of folded sheet metal attached to the beam-walls with steel strips.

The samples were adjusted to standard dimensions using electric tools, their sur-faces were polished and then they were subjected to compressive strength tests under laboratory conditions (Figure 9c). Samples were taken using diamond crowns with two diameters; *d* = 100 mm and *d* = 160 mm. The thicknesses of the precast wall and floor elements were 140 mm, so samples with *l* = 140 mm were taken. In order to determine the compressive strength, the samples with a diameter of *d* = 100 mm, were cut to *l* = 100 mm. This resulted in specimens with a ratio of *l/d* = 1. According to EN 13,791 [70], for cylindrical specimens with *l/d* = 1, the compressive strength corresponds to the strength obtained on cubic specimens with a side *a* = 15 cm. This eliminated the need for a strength correction due to the specimen dimensions. Samples of *d* = 160 mm were taken for future testing to determine Young’s modulus in compression and dynamic Young’s modulus. The calculation of the characteristic compressive strength of the concrete in the structure was also carried out in accordance with EN 13,791 [70]. The results of strength tests of core drillings taken from walls, floor slabs, and beam-walls were similar to each other. All of them were in the range of 20 to 30 MPa. According to [70], an average compressive strength of 20.5 MPa was calculated, qualifying the concrete to the current class C16/20, which is an approximate equivalent of the then class B20, according to the design assumptions. Noteworthy is the fact that the results obtained by non-destructive and destructive methods, i.e., sclerometric, ultrasound, and during testing in the machine (Figure 10) are comparable.

Drill holes were also made at the joints of the floor slabs (Figure 11c) and walls (Figure 11a,b). The quality of the floor slab and wall joints was unsatisfactory. The concrete mix was not carefully placed, and numerous cracks were found inside the concrete structure and in the contact areas of the infill with the wall dowels (Figure 11a,b). The crack opening widths of the samples were measured using Brinell magnifier. The fillings were internal elements (exposure class XC1), for which, according to EC2 [68], the crack opening width is w_k,max_ = 0.4 mm. Cracks exceeding the opening width of 0.4 mm have been classified as not fulfilling standard recommendations.

Samples taken from the floor slabs revealed missing rebar cover, which was only 3 mm in places. The pH tests of the concrete were also carried out (Figure 12). The roof slabs were characterized by advanced carbonation. The surface of the concrete was not discoloured throughout the cross-section. The effects and characteristics of carbonation as well as theoretical models to predict its depth have been described in the literature [71,72].

Phenolphthalein solution was used for preliminary pH testing. The test is carried out by spraying the substance on a fresh break of the concrete sample. At pH values higher than approximately 9, the indicator turns the concrete red-violet. Only such coloured concrete shows alkalinity sufficient to ensure passivity of reinforcement steel. Testing was conducted in accordance with EN 14,630 [73]. In the tests conducted for the textured layers of all storeys, the obtained results indicated that the layers had corroded concrete that ranged from 15 to 30 mm. In the interior walls and the structural layer of the beam-walls, the depth of carbonation ranged from 13 to 37 mm. Such deep corrosion of concrete in the interior elements is probably due to the non-usage of the building and lack of heating for over 10 years, as well as damages and defects in the window and door joinery, which affected the intensity of the aggressive environmental impact.

The level of concrete carbonatization in the texture layer was also determined in water extracts obtained by leaching the crushed concrete with distilled water. The samples taken from the core drillings were used for this purpose. Concrete was crushed in porcelain mortars, then coarse aggregate grains were removed, ground, and sieved. The sieved product was poured with distilled water and aqueous suspensions were obtained. The evaluation of the concrete’s suitability as a protective and load-bearing layer was carried out on the basis of tests using laboratory equipment with electrodes ensuring pH measurement accuracy within ±0.01. The results obtained for samples taken from the textured layer, within the pH range of 9.26–9.38, confirmed the occurrence of advanced carbonatization processes.

Part of the research work was carried out during the disassembly of individual storeys. After removal of the floor slabs, photographic documentation of the welded joints of the precast elements was made (Figure 13). Subsequently, the hidden tops of the walls were locally uncovered, and the joint sheets were cut out for laboratory testing.

Examination of several dozen specimens collected showed localized missing welds (Figure 14b), both between the tie beam and the steel plate and between the plate and the flat bars. Many of the joints had significant geometric deviations—the plates were bent on site to allow them to be installed on irregularly positioned precast wall elements. Moreover, flat bars were found to be too short and point-welded, contrary to the design’s intent. In isolated cases, flat bars were overlapped welded to another flat bar instead of to a plate embedded in the wall (Figure 14c). All plates were covered with only a superficial layer of corrosion, and it is likely that they were installed in this state when the building was constructed. The thickness of the plates, determined mechanically with a micrometer screw and by ultrasound method after cleaning from corrosion products, was in accordance with the catalog of system joints and was 6 mm. Using the ultrasound hardness tester and the correlation between steel hardness and strength, an average tensile strength of 402 MPa was estimated (Figure 14a). The steel parameters were confirmed by tensile testing in a testing machine.

## 3. Discussion. Practical Aspects of Diagnostics

The increasing age of large-panel buildings in service requires numerous assessments of their technical condition. Structural defects caused during assembly, such as inaccurate steel connections or filling of joints, are implemented in the building’s structure from the beginning of its existence. The fact that such buildings are in operation for several decades despite these defects indicates that there are large reserves of capacity and redistribution of internal forces throughout the structural system. However, structures are constantly exposed to loss of durability due to environmental aggression and material ageing. This means that in spite of the relatively good condition of the buildings and the absence of confirmed failures, the structural condition of large-panel buildings must be continuously monitored.

Demolition of such structures is rare and allows for a great number of tests, including destructive ones, which are most troublesome to the residents of the exploited structures.

In the examined construction, the authors found both execution defects and those caused by environmental aggression. Degradation that can be dangerous to the safety of the structure manifests itself by:concrete carbonation and insufficient concrete cover thickness of prefabricated elements;careless filling of the joints between the prefabricated elements;workmanship defects in welded joints;local corrosion of reinforcing steel; andexceeding of dimensional tolerance both in the prefabricated elements themselves and during assembly.

The results of ultrasound and destructive testing of vertical joints were consistent. B-scans and surface measurements of longitudinal wave velocity allowed to locate the areas of lower concrete quality, which was confirmed by examining samples obtained from core drillings. The change of concrete structure can significantly affect its material properties and it is reasonable to take it into account in terms of structural capacity.

The ultrasound and sclerometric methods for testing the quality and strength of precast concrete, were found to be consistent with the results obtained in the testing machine. It is probably correlated with the good quality of concrete, without voids, in the precast elements. The good quality of concrete was confirmed by comparing the Pulse-Echo (B-scans) and longitudinal wave velocity measurements, which clearly excluded defects in the structure of the partitions.

The lack of double-sided access to the tested elements (e.g., in the case of floors or walls without openings) limits the usefulness of the ultrasound method for precise longitudinal wave measurement. Testing is reduced to performing measurements using the Pulse-Echo method, which immediately reveals discontinuities of materials and geometric deviations. It is also helpful in locating existing installations, which is important during repair works.

The results of this study indicate that the ferromagnetic method is effective in evaluating the cover thickness and determining the distribution of reinforcing bars in precast large-panel elements, and can be helpful in selecting appropriate repair methods or locating sampling places for laboratory testing.

Testing the pH of the concrete on both fresh splits and water extracts clearly showed the carbonation processes of the concrete. For buildings in use, the least invasive approach may be to take a sample using a small-diameter diamond core and then test it under laboratory conditions, which are the most precise.

The removal of the floor slabs allowed to gain access to and assess the steel connections, which are not available for examination with current non-destructive testing equipment. The parameters of the steel used in the joints tested by correlating hardness and strength are sufficient to properly determine the strength characteristics, which was confirmed in the testing machine.

The assessment of the structure’s technical condition should be based on a comparison of the results of initial visual assessment with the results of non-destructive and destructive tests. The number of destructive tests for large prefabricated structures may be reduced to a minimum, due to cost and damage to the structure in service.

## 4. Conclusions

Demolition of a structure built of large-panel prefabricated system elements and according to system solutions is an invaluable source of knowledge about the technical condition of the examined building. It also allows to conclude on the technical condition of the whole category of similar buildings.

This paper presents a range of visual, non-destructive, and destructive tests that are helpful in determining the technical condition of reinforced concrete precast structures. On the basis of the research and correlation of the individual results, the following conclusions concerning the applied research methods of assessment of large-panel building’s degradation were drawn:to test the quality of concrete in vertical joints, it is recommended to use surface measurements of longitudinal wave velocity and B-scans;for one-sided access to the examined elements, surface scans using the Pulse-Echo method are helpful in detecting discontinuities and geometric deviations, as well as hidden installations;the quality and strength of concrete should be determined comparatively by sclerometric and ultrasound methods, and in case of discrepancies it is recommended to perform destructive tests;for testing the pH of concrete, it is recommended to take samples with small diameter diamond cores and then test under laboratory conditions;ferromagnetic testing is sufficient to determine the quality of the reinforcement work;the use of ultrasound method for crack penetration depth is helpful in monitoring crack propagation and making a possible decision to implement repair measures; anddespite the development of technology, in practice there is still a lack of equipment allowing for the assessment of the degradation of joints hidden in structural elements, such as wall joints in large-panel buildings.

The identification of defects in existing buildings with no signs of failure gives reasons for optimism about hidden capacity reserves. Nevertheless, further research is warranted to analyse their impact on structural performance. In the process of ageing of buildings, hidden safety reserves may be exhausted, especially under exceptional loads like gas explosions [29]. Therefore, it is reasonable to make maximum use of the data source, which are the few buildings to be demolished or decommissioned, or which have not been completed.

A new contribution to the knowledge of large-panel buildings will also be the analysis of the impact of hidden defects, such as carelessly made steel joints of walls, on the safety of the structure. The authors identify a research gap here and intend to perform analyses using the materials obtained from the demolition.

In order to make a reliable assessment of the condition of structural elements and their connections, it is necessary to perform static calculations of the elements that are most severely degraded. On this basis, further decisions related to the repair, reinforcement or demolition of the structure should be made. Actual values of physical and strength parameters of materials exploited for many years should be used for calculations. These parameters can be obtained from non-destructive testing.

## Figures and Tables

**Figure 1 materials-15-03770-f001:**
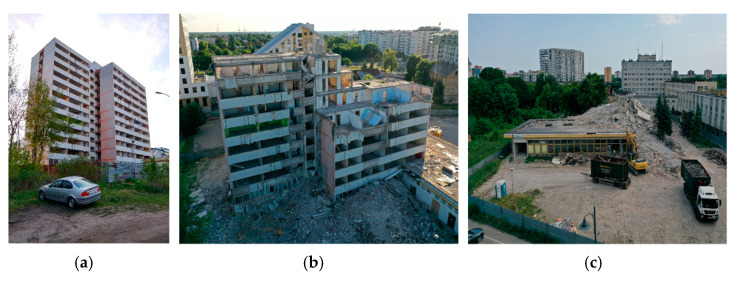
View on the analysed building: (**a**) before demolition; (**b**) after demolition of top storeys; and (**c**) in the last stage of demolition.

**Figure 2 materials-15-03770-f002:**
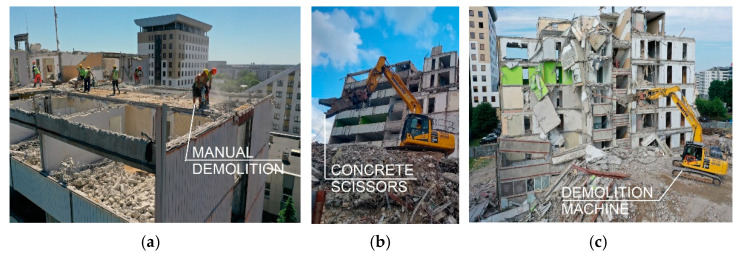
Demolition of the building: (**a**) manual demolition; and (**b**,**c**) mechanical demolition.

**Figure 3 materials-15-03770-f003:**
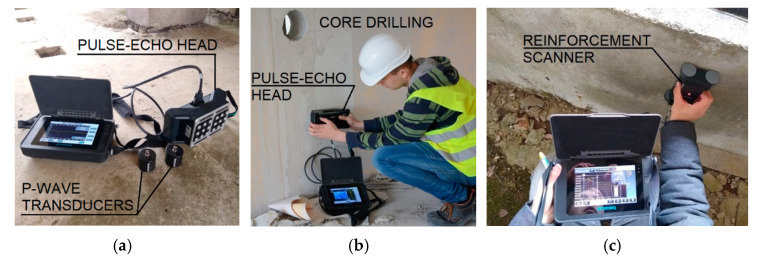
NDT tests: (**a**) ultrasound testing equipment; (**b**) vertical joint scan; and (**c**) ferromagnetic testing.

**Figure 4 materials-15-03770-f004:**
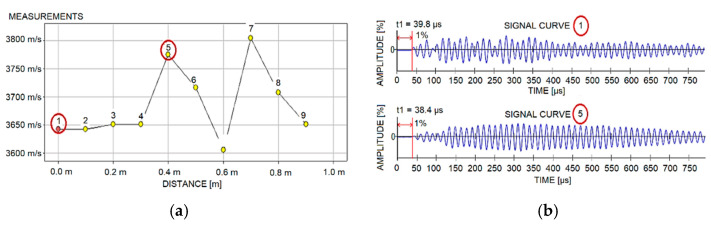
Ultrasound tests: (**a**) P-wave measurement graph at 10 points; and (**b**) P-wave propagation graph at points 1 and 5.

**Figure 5 materials-15-03770-f005:**
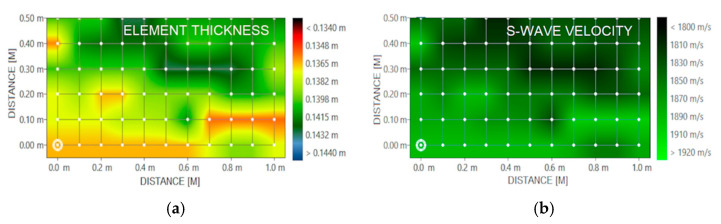
Area scans: (**a**) element thickness; and (**b**) S-wave velocity.

**Figure 6 materials-15-03770-f006:**
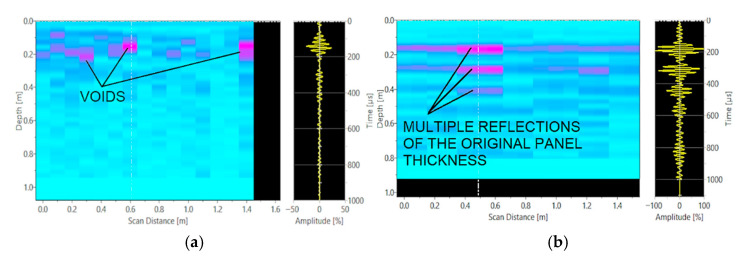
B-scans: (**a**) vertical wall joint; and (**b**) walls.

**Figure 7 materials-15-03770-f007:**
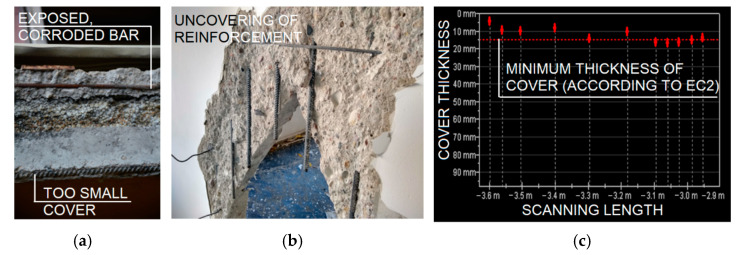
Determining the quality of reinforcement work: (**a**) corroded rebar and lack of cover; (**b**) uncovering of reinforcement; and (**c**) ferromagnetic scan—distribution of reinforcement and thickness of cover.

**Figure 8 materials-15-03770-f008:**
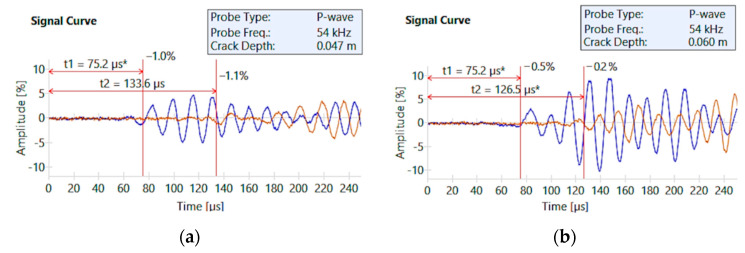
Ultrasound measurement of the crack depth: (**a**) crack in the longitudinal wall; and (**b**) crack in the transverse wall.

**Figure 9 materials-15-03770-f009:**
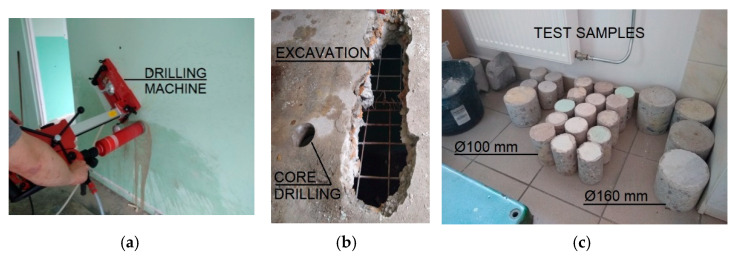
Drilled cores and excavations: (**a**) core-drilling machine; (**b**) view of excavations and drill core in floor slab; and (**c**) samples in the laboratory.

**Figure 10 materials-15-03770-f010:**
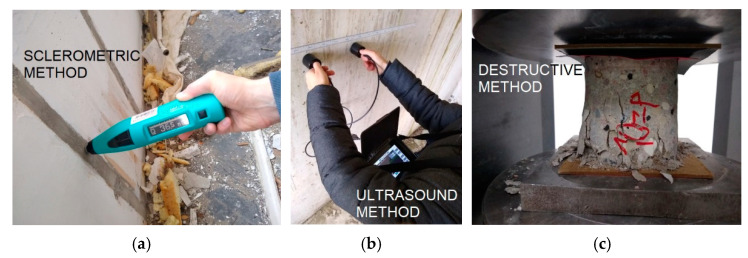
Concrete strength testing: (**a**) sclerometric method; (**b**) ultrasound method; and (**c**) destructive method.

**Figure 11 materials-15-03770-f011:**
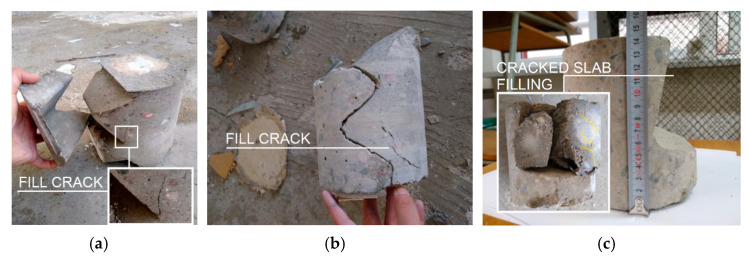
Joints core drilling: (**a**,**b**) core drilling of fill with visible cracks; and (**c**) floor slab joint.

**Figure 12 materials-15-03770-f012:**
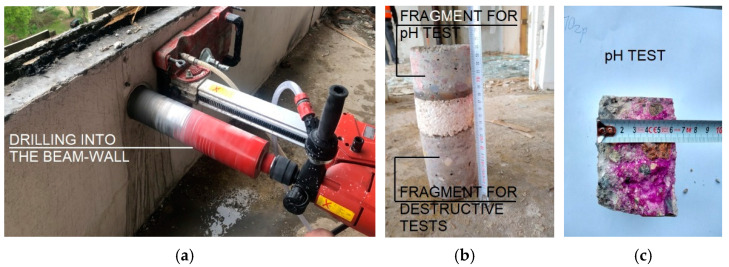
Beam-wall tests: (**a**) core drilling; (**b**) sample for testing; and (**c**) pH test.

**Figure 13 materials-15-03770-f013:**
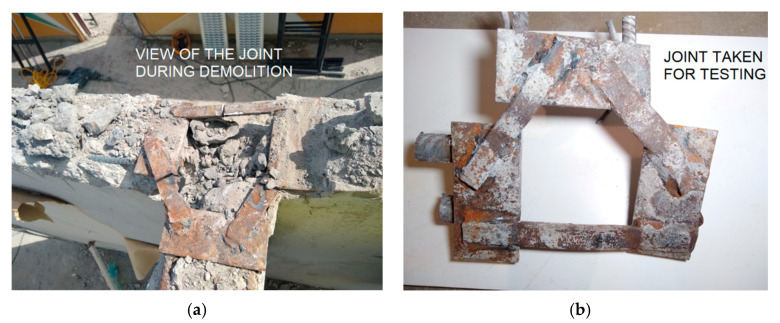
Steel joint 3 internal walls: (**a**) in the process of demolition; and (**b**) sample taken for testing.

**Figure 14 materials-15-03770-f014:**
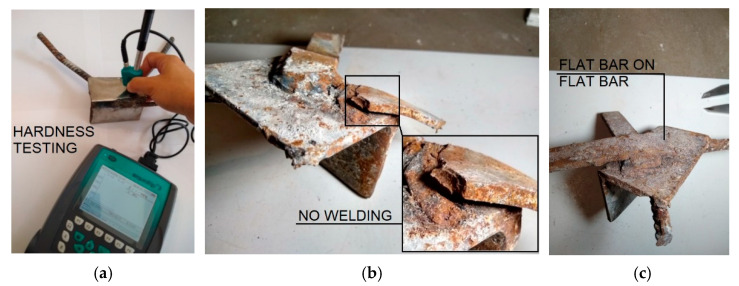
Connection sheets: (**a**) hardness testing of steel; (**b**) missing weld; and (**c**) incorrectly made joint.

## Data Availability

The data presented in this study are available on request from the corresponding author.

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
