# Peer review of "Degradation Analyses of Systemic Large-Panel Buildings Using Comparative Testing during Demolition"

_materials, 2022, doi:10.3390/ma15113770_

Round 1

Reviewer 1 Report

The topic of this paper is related to the degradation analyses of systemic large-panel buildings using comparative testing during demolition. This research aims to concentrate on determining the research methods for evaluating the degradation state of structural elements of prefabricated buildings. The approach focuses on determining the technical condition of prefabricated buildings by visual, non-destructive, and destructive tests of an over 40-year-old large-panel building scheduled for demolition. This paper provides some valuable methods, results, and solutions for determining the technical condition of reinforced concrete precast structures. I recommend this paper can be revised according to clarify the following issues:

  1. For all Figures shown in this paper, please make sure there is one line spacing before and after each figure.
  2. Page 5, lines 181 and 182. Please explain why choosing 10-15 cm intervals as measurement points and why 5 to 10 readings were taken on each wall.
  3. Page 5, Figure 4, the resolution of this Figure is a little low. Please re-edit it in Excel and insert it into Microsoft Word.
  4. Page 5, Figure 5, the unit of 0.134 and 0.144 of the color bar in Figure 5(a) is not shown clearly. Please re-edit it. On the other hand, the scanned area (1m x 0.5m) is only part of the wall, is that right? Please also make sure the area is shown in Figure 5(a) and Figure 5(b) is the same area.
  5. Page 6, Figure 6, lines 215 and 216. The authors state pink and purple colors mean defects or discontinuities. Please explain a little bit about what the darker (purple) or lighter (pink) color means? Are the darker color means severe defects?
  6. Page 6, Figure 7(c), please describe a little bit about Figure 7(c). What is the yellow dashed line means? What are the gray points in this Figure mean? What is the title of the x-axis and y-axis of Figure 7(c)? Please explain a little bit about line scans and surface scans in lines 221 and 222, what these tools are used for and what the difference is. On the other hand, it seems that Figure 7(c) lacks one part in the x-axis direction. Please make sure the whole Figure is plotted and increase the resolution if possible.
  7. Page 7, Figure 8, please make sure the font size of words shown in each Figure maintains the same. For Figure 8(c), what are the sizes of concrete cylinder samples? It seems that there are two different sizes of samples, and what standards do the authors follow to choose and test concrete cylinders.
  8. Page 8, Figure 10, did the authors measure the crack width of drilling samples? How do the authors define the visible cracks, and what crack width will cause the structure or samples to be categorized as unsatisfactory?
  9. Page 8, Figure 11(c), please make a short description for Figure 11(c) because most of the researchers who work in the structural (not material) scale may not be very familiar with this test.

Finally, I will give two suggestions about all the figures shown in this paper. The resolution is also an essential factor in improving the paper acceptance rate for a high-quality paper. (1) When plotting a Figure and adding notation (words, numbers) on an original paper, it is better for the figure can be plotted in Microsoft Excel and edited in Microsoft software (PowerPoint or Word) and group them. Then, insert it into Microsoft Word. In this way, the resolution of the notation shown on each picture will have the same resolution as the words in this paper. Please also make sure the notations shown in each Figure have the same size (width and height). (2) Please check the minor edit and grammar errors shown in this paper.

Reviewer 2 Report

It is clear that the authors have conducted a fascinating series of investigative experiments. The writing quality, on the other hand, is poor. There are several incorrect sentences, and there is a distinct lack of flow between sentences and paragraphs, making it difficult to understand what the author was trying to convey. It is not yet ready for publication in a high-impact journal, and I urge the authors to rewrite the manuscript and improve its readability.

Reviewer 3 Report

Manuscript No.: materials-1713907

Title: Degradation analyses of systemic large-panel buildings using comparative testing during demolition

Suggestion: Major revision

Review: This paper presented visual, non-destructive and destructive tests of an over 40-year-old large-panel building scheduled for demolition. The following comments are noticed during the reviewing:

Detailed comments:

  1. More detailed results from these tests should be presented other than only posting the related test instruments and test methods.
  2. The authors are encouraged to present the new findings in their work compared with the previous studies. 
  3. More recent literature on fracture properties of concrete should be discussed and compared, such as the work on sludge ceramsite concrete, and the role of silica fume and steel fibre on recycled aggregate concrete from Prof. Jianhe Xie. And the work on waste seashells by Prof. Engui Liu.
  4. Also, in the introduction part, the authors should highligh the durability of concrete, by refering more related literatures, such as the chloride diffusion and migration cofficients of cracks in concrete. To point out the meanings of realted testing including the cracks and other important parameters.
  5. If possible, please provide any microstructure results from lab tests, such as SEM, MIP.

Round 2

Reviewer 2 Report

The authors attempted to address the provided comments, and the paper has improved somewhat but still requires improvement. 

The work focuses on determining the technical condition of large-panel buildings using all possible methods including visual, nondestructive and destructive testing. So, there are no problems regarding the research approach.

As mentioned in my review, the main problem with this paper is its readability. In fact, the revised version is much better than the original manuscript. However, there are still several sentences that need improvement to make them clearer. Just to give you an example, “Due to location of the building in the very centre of the city and the proximity to the adjacent building, it was decided that the demolition would be carried out in a way that would minimize inconvenience to neighbouring structures and their users.” This sentence doesn’t read correctly. It can be rephrased as “Due to the building's location in the city center and its proximity to the adjacent building, it was decided to demolish in a way that minimized the inconvenience to neighboring buildings and their occupants.”

Author Response

Dear Reviewer,

Thank you for appreciating the corrections we made to our article. Thanks to the reviewer's earlier suggestions, the quality of the manuscript has improved. We have sent the text to a native English speaker who has helped to rephrase the article to improve its readability. We hope that thanks to the introduced changes, the text will meet the expectations of the reviewer and will be easy to read in an international environment.

Reviewer 3 Report

The paper has been revised accordingly.

Author Response

Dear Reviewer,

Thank you for appreciating the corrections we made to our article. Thanks to the reviewer's earlier suggestions, the quality of the manuscript has improved.